# FlexParallel: Automatic Parallelism Tuner via Grey-Box Optimization for Training Giant Models

## Abstract

The rapid scaling of large language models (LLMs) has elevated parallel configuration tuning to a central challenge. Most existing frameworks rely on labor-intensive manual tuning. While recent advances attempt to automate this process and reduce reliance on expert intervention, their effectiveness often depends on highly accurate cost models. In practice, such models frequently fall short due to the challenge in exact modeling, leading to suboptimal configurations. To address the limitation, this work introduces *FlexParallel*, a framework that integrates an uncertain-aware grey-box cost surrogate, a sample-efficient parallelism explorer, and an adaptive stopping criteria, to automatically discover high-performance parallelism configuration. We evaluated the effectiveness of FlexParallel through extensive experiments spanning diverse model architectures, parameter scales, sequence lengths, and cluster sizes. To our best knowledge, this work presents the first empirical evaluation of automatic parallelism tuner on a cluster of up to 8,192 devices. Experimental results demonstrate that, with high sample efficiency, Flex-Parallel achieves an average speedup of $1.06\times$ over manual expert tuning, and up to $1.12\times$ in the best case.

## 1 Introduction

Recent advances in transformer-based large language models (LLMs) have demonstrated that scaling up both model parameters alongside training data volume markedly improves model capabilities, giving rise to emergent abilities such as instruction following and complex reasoning (Achiam et al., 2023; Touvron et al., 2023a; Chowdhery et al., 2023; Wei et al., 2022). Nevertheless, the pursuit of ever-larger models and longer contexts incurs immense computational and memory demands, which far exceed the capabilities of individual computing devices. This challenge has spurred considerable interest in distributed training techniques across academia and industry.

Substantial research has focused on developing parallelism methods. Among them, data parallelism (DP) (Li et al., 2020) divides training data into mini-batches, with each device processing a distinct batch to compute model gradients, and multiple devices synchronizing the gradients periodically. While DP enhances training scalability, it necessitates a complete replica of the model on each device. By contrast, pipeline parallelism (PP) (Huang et al., 2019; Narayanan et al., 2019) segments the model itself into sequential stages, and distributes them across multiple devices. Because forward and backward computations are organized in a pipelined manner, only the activations and their gradients at stage boundaries need to be communicated between adjacent stages. This results in a relatively low communication overhead. However, the need to store all intermediate activations until their backward pass is completed still incurs considerable memory consumption. Tensor parallelism (TP) (Shoeybi et al., 2019), on the other hand, partitions the weights of individual model layers across devices, using collective communication to aggregate their computation results. Such a partition alleviates both static memory and activation memory consumption, but is accompanied by increased communication overhead, especially as the number of devices increases.

Given their respective strengths and weaknesses, mainstream distributed training frameworks—such as Megatron (Shoeybi et al., 2019), DeepSpeed (Rasley et al., 2020), and MindSpeed (Ascend, 2025a)—support all major parallelism techniques. However, as model sizes and training clusters

continue to scale up, manually configuring hybrid parallelism requires substantial engineering effort and deep expertise in both machine learning and systems infrastructure (Narayanan et al., 2021). The intricate interactions among different parallelism methods further complicate the search for optimal configurations, making it highly challenging even for domain specialists. These challenges have spurred interest in automatic tuning methods for parallelism configuration (Liang et al., 2023; Zheng et al., 2022).

Existing studies for automatic parallelism tuning typically adopt a two-stage framework (Miao et al., 2022; Zheng et al., 2022; Liu et al., 2023a): (1) constructing a cost model to predict per-iteration runtimes for candidate configurations; and (2) employing a solver to search the combinatorial space for final configuration recommendation, guided by the cost model. Nevertheless, accurately modeling runtimes in practice remains a major challenge.

For instance, consider multiplying matrices $A \in \mathbb{R}^{m \times n}$ and $B \in \mathbb{R}^{n \times k}$, a fundamental operator in LLMs. Although the theoretical number of floating-point operations required for this computation is approximately $2mnk$, the actual runtime is strongly affected by hardware and software characteristics. These include the number of tensor cores, L1/L2 cache sizes, memory bandwidth, and the tiling and scheduling strategies. In addition, modern hardware platforms often provide vendor-optimized libraries that are tailored to specific matrix shapes, leading to pronounced variation in computational efficiency across different input shapes (Fatahalian et al., 2004; Narayanan et al., 2021). This variability is illustrated in Figure 1. Consequently, analytical cost models frequently exhibit discrepancies in performance rankings between predictions and actual measurements. These inaccuracies can further result in suboptimal or even infeasible configuration recommendations.

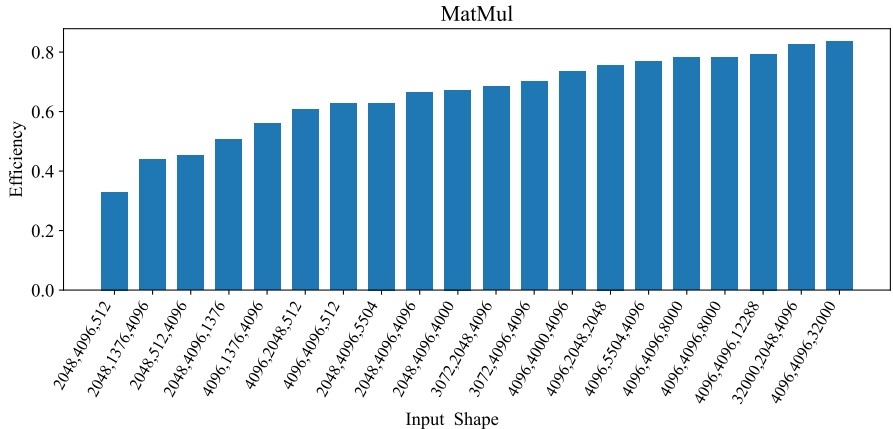

Figure 1: Fluctuation of computational efficiency across input shapes $(m, n, k)$. The efficiency is computed via $\frac{2mnk}{t \times \text{FLOPS}}$, where $t$ denotes the measured runtime, and FLOPS denotes floating-point operations per second of the considered device.

To tackle this challenge, we introduce *FlexParallel*, an automatic parallelism tuner designed to account for the inherent inexactness of cost estimation, and proactively collect feedback for continuous refinement of configuration recommendation. As illustrated in Figure 2, FlexParallel consists of three key components: (1) an uncertainty-aware cost surrogate, (2) an efficient parallelism explorer and (3) an adaptive stopping criteria. The cost surrogate models a full distribution over per-iteration runtime for each parallelism configuration, capturing not only predicted performance but also prediction confidence. With these information, the explorer identifies a potential configuration candidate, which may show low predicted performance yet be plausibly undervalued due to the limited model knowledge. By temporarily applying the selected configuration in LLM training for several iterations, FlexParallel gathers actual runtime data and use it to further refine the cost models. This process of configuration exploration and cost model update is executed alternatively, until the stopping criteria is met, which is designed to ensure with high probability that no superior configuration remains unexplored. Ultimately, the configuration yielding best performance during exploration is adopted as the final recommendation.

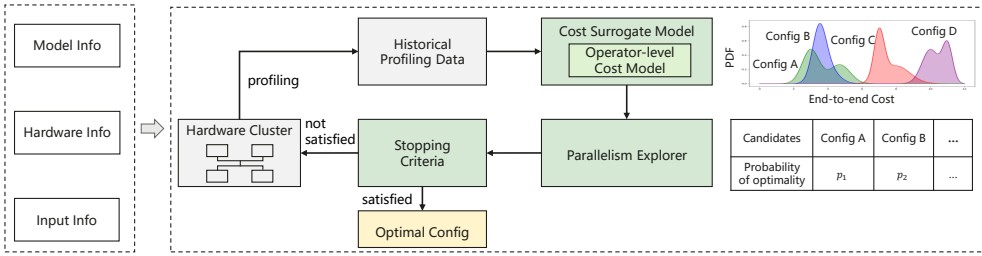

Figure 2: System overview of FlexParallel.

To assess the effectiveness of FlexParallel, we conduct extensive experiments across a diverse set of LLMs under various conditions. These tests cover both dense and MoE architectures, parameter scales ranging from 7B to 135B, cluster sizes from 8 to 8192 NPUs, and sequence lengths from 2K to 32K tokens. The results show that FlexParallel delivers an average training performance improvement of 6%, and up to 12% in the best case, relative to manual expert tuning, while requiring the exploration of at most 2% of the total configuration space before termination.

## 2 RELATED WORKS

**Automatic Parallelism Tuner.** Manual design of parallelization configurations is a relatively coarse-grained optimization process. Enabling fine-grained optimization for parallelization configurations motivates recent work on automatic solver. The primary challenges in designing tractable automatic solvers stem from the gigantic search space and the difficulty in developing accurate cost models. PP incurs high memory consumption, TP requires frequent synchronous collective communication, and SP heavily depends on the input sequence length for effective scaling. The integration of hybrid parallelism strategies (e.g., TP, PP, DP, SP, UP) introduces intricate interdependencies between computational operators and collective communication operations. Due to the search space complexity, lots of related works only consider the combination of a few parallelization methods, but not all of them (Fan et al., 2021; Jia et al., 2019; Narayanan et al., 2019; Wang et al., 2019). For example, (Lai et al., 2023) designs Merak to automatically orchestrate DP, PP and TP strategies. Galvatron (Miao et al., 2022) formulates the search for optimal hybrid parallelism (PP, DP, TP) as a dynamic programming problem, incorporating expert knowledge to prune infeasible options. Alpa (Zheng et al., 2022) is a pioneering framework that takes all parallelization methods into consideration. Colossal-Auto (Liu et al., 2023a) jointly optimizes operator sharding strategies and activation recomputation strategy, balancing memory and computational efficiency. The aforementioned automatic solvers for parallelization configurations require access to accurate cost models. Based on the cost model, they can solve a near-optimal configuration via mixed-integer linear programming or heuristics such as dynamic programming.

## 3 METHOD

To facilitate efficient parallelism tuning, the component design of FlexParallel must overcome several challenges. First, the relationship between parallelism configuration and per-iteration runtime is complex and highly task-dependent. As a result, training naive black-box models not only suffers from low data efficiency, but also requires retraining separate models for each task. Second, the distribution of per-iteration runtime often lacks a closed-form expression, making it infeasible to be directly utilized in the design of exploration strategies and termination mechanisms. To overcome these obstacles, Section 3.1 leverages architecture knowledge of LLM and integrates simulated performance data to enhance the learning and prediction of the probabilistic cost surrogate. Sections 3.2 and 3.3 employ a sample-based exploration strategy and a corresponding termination criterion respectively, thereby avoiding the need to explicitly evaluate the entire runtime distribution. These two designs constitute the core methodology of FlexParallel.

## 3.1 Cost Surrogate Model

Noting that transformer-based LLMs share a common set of core operators. This means that once we establish cost models for these operators, they can be reused across different tasks. This motivates to aggregate operator-level models to construct the cost surrogate in FlexParallel. Particularly, the overall cost of given parallelism configuration and LLM is computed as:

$$\text{cost}(\text{config}, \text{LLM}) = u\bigg(\Big\{ g_i\big(f_i(\text{config}, \text{LLM})\big) \Big\}_{i=1}^{I}, \text{LLM}\bigg), \tag{1}$$

where $i$ denotes the index of operators within the LLM, $f_i$ determines the shape of input tensors for the operator, $g_i$ estimates the runtime of individual operator given the input shape, $u$ synthesizes the end-to-end runtime by considering the dependencies among the operators.

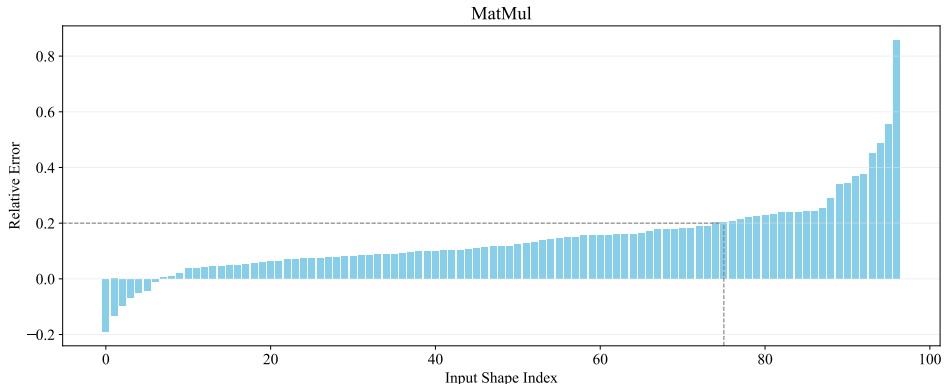

Figure 3: Distribution of relative error between simulated and profiling-based runtimes for the Mat-Mul operator across different tensor shapes.

In practice, both $f_i$ and $u$ are explicit rules that can be automated within a workflow. Thus, our primary focus is modeling $g_i$. To improve learning efficiency and support robust extrapolation of $g_i$, we incorporate simulated operator performance as an auxiliary signal during both the training and prediction phases. Specifically, the simulated performance is measured by applying the operator to arbitrary inputs with a target shape. Although simulated data is less faithful than that obtained from real LLM training traces, it provides a reasonable initial estimate at a negligible cost.

Figure 3 illustrates the relative errors between simulated and profiling-based runtimes for MatMul across a variety of input shapes. Notably, for 75% of the cases, the relative error remains below 20%. Considering this correlation, we develop operator-level models to predict the relative error. Subsequently, the final runtime estimation is obtained by correcting the simulated value with the predicted error, as illustrated in Figure 4 and detailed below.

To elaborate, we focus on a single operator. Let $x_i = f_i(\text{config}, \text{LLM})$ denote the operator's input shape, and $t_i = g_i(x_i)$ its corresponding runtime during LLM training. Historical profiling constitutes the dataset $\big\{\big(x_i^{(n)}, t_i^{(n)}\big)\big\}_{n=1}^{N_i}$. For any $x_i$, we collect the simulated runtime, denoted by $\hat{t}_i$. Then, the operator-level cost model aims to learn a probabilistic mapping from $x_i$ to the relative error

$$e_i = \frac{t_i - \hat{t}_i}{\hat{t}_i}. \tag{2}$$

This is implemented by training a Gaussian Process (GP) model for each considered operator.

During prediction, when presented with an input of shape $x_j$, we first conduct the operator simulation to produce an initial estimate $\hat{t}_j$. Next, the GP predicts a distribution $p_i(e_j|x_j)$ for the relative error $e_j$, from which the cost is refined as $t_j = (1 + e_j)\hat{t}_j$. Because $e_j$ is a random variable, its stochasticity propagates to both the operator-level and the end-to-end cost estimates, rendering them random variables as well. We denote the resulting end-to-end cost distribution by $p(\text{cost} \mid \text{config}, \text{LLM})$. The concrete form of this distribution is determined by the aggregation

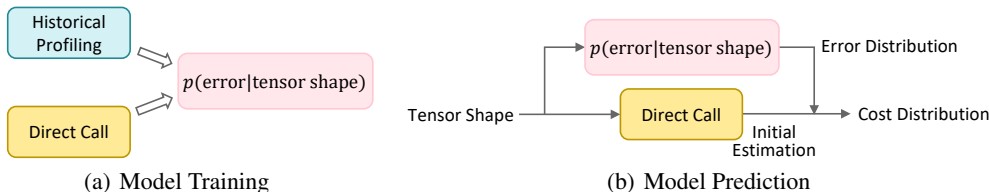

(a) Model Training  (b) Model Prediction

Figure 4: Operator-level Surrogate Model.

of operator costs into the end-to-end cost, as described in equation 1. Owing to the intractability of deriving the closed form, we do not address its explicit computation here. Nonetheless, it is important to note that this distribution encodes information about both the performance prediction and the associated uncertainty. These characteristics are pivotal for efficient exploration and timely termination, as will be discussed in the next subsection.

## 3.2 PARALLELISM EXPLORER

Ideally, once the end-to-end cost distribution $p(\text{cost} \mid \text{config}, \text{LLM})$ is determined for each parallelism configuration, various algorithms can be leveraged to identify the most promising configuration. For example, the Upper Confidence Bound (UCB) Auer (2002) algorithm selects the configuration according to

$$\min_{\text{config}} : \text{Mean}\Big(p(\text{cost} \mid \text{config}, \text{LLM})\Big) - \beta \cdot \text{Std}\Big(p(\text{cost} \mid \text{config}, \text{LLM})\Big), \qquad (3)$$

where $\beta$ is a tunable hyperparameter balancing exploitation and exploration. The mean term captures the average predicted cost for a configuration, favoring those with lower estimated runtimes. The standard deviation term reflects epistemic uncertainty in model's predictions, highlighting configurations for which the cost remains ambiguous due to limited data. Prioritizing configurations with both low predicted cost and high uncertainty encourages the parallelism tuner to gather informative measurements in unexplored regions of the configuration space, thus improving the accuracy of the surrogate model and the quality of subsequent recommendations.

The objective function in equation 3 clearly outlines two key factors to consider. However, it is not applicable in our setting, due to the intractability of deriving an explicit expression for the end-to-end cost distribution. Instead, FlexParallel adopts a sample-based exploration strategy, which extends Thompson sampling (Russo et al., 2018) by placing more emphasis on exploration over exploitation.

Let $\text{config}^* = \arg\min_{\text{config}} \text{cost}(\text{config}, \text{LLM})$ denote the optimal configuration. Importantly, $\text{config}^*$ is itself a random variable governed by the underlying distribution $p(\text{config}^* \mid \text{LLM})$. Thompson sampling explores the configuration space by sampling from $p(\text{config}^* \mid \text{LLM})$, which amounts to computing

$$\widetilde{\text{config}}^* = \arg\min_{\text{config}} \widetilde{\text{cost}}(\text{config}, \text{LLM}), \qquad (4)$$

where the tilde indicates a sample from the corresponding random variable. The sample of end-to-end cost, $\widetilde{\text{cost}}$, is generated through the following procedure:

$$\widetilde{\text{cost}}(\text{config}, \text{LLM}) = u\Big(\{\tilde{t}_i\}_{i=1}^{I}, \text{LLM}\Big), \qquad (5a)$$

$$\tilde{t}_i = (1 + \tilde{e}_i)\hat{t}_i, \qquad (5b)$$

$$\tilde{e}_i \sim p_i(e_i \mid x_i). \qquad (5c)$$

Although $p(\text{cost} \mid \text{config}, \text{LLM})$ is not explicitly maintained, both its mean prediction and uncertainty intrinsically influence the sampling process. As shown in Figure 5, Config A exhibits a higher mean cost but also greater uncertainty, compared to Config B. Consequently, there remains a non-negligible chance that $\widetilde{\text{cost}}(\text{config}_A, \text{LLM}) < \widetilde{\text{cost}}(\text{config}_B, \text{LLM})$, prompting the exploration of configuration A. In contrast, configurations C and D exhibit significantly higher costs than A and B. Despite respective uncertainties, the likelihood that their costs drop below those of A or B is almost zero, and thus these configurations are unlikely to be selected for exploration.

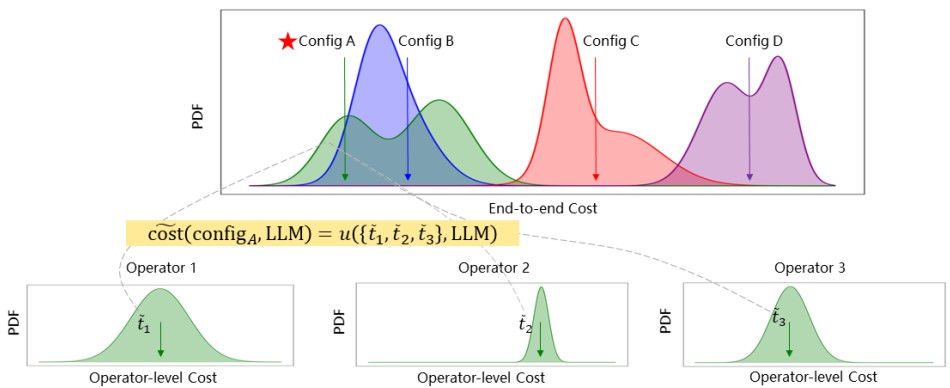

Figure 5: End-to-end Cost Distribution and Its Empirical Sample.

In standard Thompson sampling, the selected configuration will be applied for several iterations of LLM training, after which profiling traces are gathered to update the surrogate cost model. Repeating this process minimizes the long-term cumulative cost from an information-theoretic perspective. However, each switch in parallelism configuration imposes additional overhead, primarily due to LLM resharding and cluster-wide training initialization. To reduce this overhead, FlexParallel attempts to identify the most efficient configuration with minimal exploration and then employs this configuration throughout the full-scale LLM training. A key distinction for this goal is that we should avoid re-exploring configurations that have already been tried. Although re-executing a previously well-performed configuration incurs low immediate cost, it yields little additional information for refining the surrogate model. To prevent it, we incorporate a simple modification to standard Thompson sampling.

Specifically, we maintain a set $\mathcal{E}$ to record all configurations that have already been explored. At each exploration step, rather than sampling a single configuration, we generate a batch of $M$ candidates, denoted as the sequence $\mathbb{S} = \left(\text{config}^*(i)\right)_{i=1}^{M}$. After filtering out explored configurations to form $\bar{\mathbb{S}} = \left(\text{config}^*(i)\right)_{i \in \mathcal{I}}$, $\mathcal{I} = \{i | \text{config}^*(i) \notin \mathcal{E}\}$, we sample uniformly at random from $\bar{\mathbb{S}}$ for exploration.

### 3.3 STOPPING CRITERIA

The empirical population $\mathbb{S}$ also provides a more comprehensive characterization of $p(\text{config}^* \mid \text{LLM})$ than relying on a single sample. By utilizing both $\bar{\mathbb{S}}$ and $\mathbb{S}$, we can estimate the probability that the optimal configuration has already been explored as:

$$p^* = \frac{|\mathbb{S}| - |\bar{\mathbb{S}}|}{|\mathbb{S}|}. \tag{6}$$

As $p^*$ approaches 1, the likelihood that the optimal configuration remains unexplored becomes negligible. To avoid incurring unnecessary overhead from continued exploration, we introduce the following stopping criterion: exploration is terminated once $p^* > \eta$. The predetermined threshold $\eta$ balances the exploration overhead against the probability of missing the optimal configuration.

The integration of the cost surrogate, parallelism explorer, and stopping criteria forms the core of FlexParallel, with the full procedure summarized in Algorithm 1.

## 4 EXPERIMENTS

In this section, we evaluate the effectiveness of FlexParallel by examining three questions:

- **Rank Preservation**: To what extent do predicted runtimes preserve the ordering of the true runtimes across configurations?

---

**Algorithm 1** FlexParallel

---

**Require:** Exploration budget $T$, historical profiling $\mathcal{D}_i = \left\{ \left( x_i^{(n)}, t_i^{(n)} \right) \right\}_{n=1}^{N_i}$ for each operator.

1: Let $\mathcal{E} = \emptyset$.
2: **for** $t = 1$ **to** $T$ **do**
3:     Acquire simulated performance data $\widehat{\mathcal{D}}_i = \left\{ \left( x_i^{(n)}, \hat{t}_i^{(n)} \right) \right\}_{n=1}^{N_i}$ for each operator.
4:     Compute the relative error $e_i^{(n)}$ according to equation 2, and train the GP model using $x_i^{(n)}$ as input and $e_i^{(n)}$ as supervised signal.
5:     Draw $M$ samples according to equations 4 and 5 to constitute the empirical population $\mathbb{S}$.
6:     Compute the probability $p^*$ according to equation 6.
7:     **if** $p^* > \eta$ **then**
8:         **return** $\arg\min_{\text{config} \in \mathcal{E}} \text{cost}(\text{config})$.
9:     **else**
10:        Randomly pick a configuration $\text{config}_t$ from $\bar{\mathbb{S}}$ for several iterations of LLM training, record the per-iteration runtime as $\text{cost}(\text{config}_t)$, and collect the corresponding profiling data as $\bar{\mathcal{D}}_i$.
11:     **end if**
12:     Let $\mathcal{D}_i = \mathcal{D}_i \cup \bar{\mathcal{D}}_i$ and $\mathcal{E} = \mathcal{E} \cup \left\{ \text{config}_t \right\}$.
13: **end for**

---

- **Exploration Efficiency**: How many queries does FlexParallel require to discover the optimal configuration?

- **Empirical Validation**: What performance gains does FlexParallel achieve compared to expert-designed manual tuning across a variety of mainstream models?

## 4.1 EXPERIMENTAL SETUPS

We tune micro batch size (MBS) and the degree of seven parallelism methods—TP, DP, PP, VPP, CP, UP and EP. All experiments are conducted on Ascend clusters, where each server is equipped with 8 NPUs, each NPU providing 64GB of memory. The 8 NPUs within a node are interconnected via high-speed mesh topology links. Experiments are implemented using Python 3.8 and PyTorch 2.1.0. For distributed training, we use MindSpeed (Ascend, 2025a) and MindSpeed-LLM (Ascend, 2025b), both of which provide efficient parallelism support for LLM training on NPUs.

For benchmarking, we considered a variety of transformer-based LLMs, including GPT Radford et al. (2019), Qwen 2.5 (Yang et al., 2024), LLaMA 1/2/3 (Touvron et al., 2023b; Dubey et al., 2024), Mixtral MOE (Jiang et al., 2024), and Pangu (Yin et al., 2025), with detailed model and training configuration summarized in Table 1. When implementing FlexParallel, we build operator-level surrogate models for a representative set of operators, including MatMul, BatchMatMul, Flash Attention, LayerNorm and RmsNorm.

Table 1: Model and training configuration.

| Model | # of Layers | Hidden Size | Seq. Length | Global Batch Size |
|---|---|---|---|---|
| LLaMA-7B | 32 | 4096 | 2048 | 256 |
| LLaMA-13B | 40 | 5120 | 2048 | 128 |
| LLaMA2-13B | 40 | 5120 | 4096 | 512 |
| LLaMA2-34B | 48 | 8192 | 4096 | 1024 |
| LLaMA3-8B | 32 | 4096 | 8192 | 64 |
| GPT-13B | 40 | 5120 | 4096 | 128 |
| GPT-38B | 48 | 8192 | 4096/32768 | 128 |
| Qwen2.5-7B | 28 | 3584 | 8192 | 64 |
| Pangu 135B | 94 | 12288 | 4096 | 1536 |
| Mixtral $8 \times 7B$ | 32 | 4096 | 32768 | 32 |

## 4.2 EFFECTIVENESS OF COST SURROGATE MODEL

To validate our cost model, we evaluate how well its predicted runtimes rank against the ground-truth performance across the entire configuration space. We quantify this rank correlation using the Kendall tau coefficient $\tau$, where a value approaching 1 indicates a near-perfect alignment. Ground-truth execution times were measured by running the LLaMA-7B model for each configuration on an 8-NPU system. Our baseline is Calculon (Isaev et al., 2023), a state-of-the-art framework that estimates training performance with white-box modeling. Notably, our cost model achieves $\tau = 0.86$, a significant improvement over the Calculon's baseline $\tau = 0.76$, which underscores its superior rank preservation. Detailed results are provided in Appendix A.2.

## 4.3 EFFICIENCY OF CONFIGURATION TUNING

Our approach dynamically refines its cost model by integrating exploration information at each step. In each iteration, we draw $M = 100$ samples from the cost model to form the empirical population $\mathbb{S}$. A configuration's probability of being optimal is its frequency within sample $\mathbb{S}$. Figure 6 illustrates the evolution of probability distribution of the optimal configuration $\text{config}^*$ during the search process. As shown in Figure 6 (a), before exploration, four candidate configurations were initially identified as potential optima based on low-fidelity data. By the second iteration, the search narrows to just two candidates. The exploration process converged and terminated after four iterations, once the stopping criteria $\eta \geq 0.95$ is met. Crucially, the final configuration identified by our method matches the true optimum found via an exhaustive search of the entire configuration space.

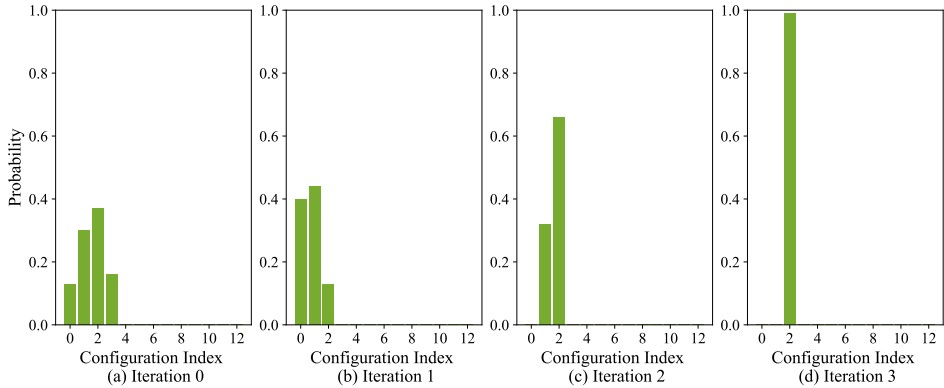

Figure 6: Evolution of probability distributions of the optimal configuration $\text{config}^*$ during the search process.

## 4.4 PERFORMANCE ANALYSIS

We adopt the expert configurations from the MindSpeed-LLM framework as baselines. These expert-curated configurations are carefully optimized to maximize training throughput. Table 2 shows that FlexParallel achieves an average of $1.06\times$ (up to $1.12\times$) speedup for 10 mainstream LLMs compared to manual expert tuning. For example, when training a GPT-38B model with a long sequence length of $32,678$ tokens, FlexParallel reduces the end-to-end training time by $12\%$. Notably, our auto-tuner identifies the optimal configuration with high sample efficiency, evaluating fewer than $2\%$ of the total search space candidates.

To the best of our knowledge, this work represents the first empirical validation of an auto-tuning algorithm on a cluster with 8192 NPUs for distributed training. FlexParallel consistently identifies the optimal configurations for the Pangu-135B model as the cluster grows from 6144 to 8192 NPUs. At the 6144 NPU scale, FlexParallel achieves a $5\%$ performance gain. Experts typically set $TP = 8, PP = 8$ to ensure that the partitioned model fits within a single NPU's memory. This leaves a data parallelism degree $DP = 96$. Because the communication domain size (96) is not a power of two, this choice degrades the performance of All-Reduce operations. FlexParallel identifies a

superior configuration of $PP = 6, DP = 128$, which reduces pipeline bubbles and improves the data-parallel communication bandwidth.

FlexParallel excels at identifying parallelism strategies that effectively navigate the trade-offs between computation, memory, and communication. Taking the GPT-13B model as an example, experts typically employ $TP = 8$ to reduce both static and dynamic memory consumption, while increasing MBS to $4$ for higher computational efficiency. In contrast, FlexParallel discovers an optimal configuration ($PP = 8, MBS = 1, TP = 1$), thereby avoiding heavy TP communications. Our analysis reveals that even models within the same architectural series require different parallelism configurations for optimal performance. For example, FlexParallel found the best configuration for LLaMA-13B to be $PP = 8, VPP = 5, MBS = 1$. However, for LLaMA2-13B, the optimal setup was entirely different: $TP = 8, MBS = 4$.

The results demonstrate that FlexParallel automatically discovers optimal parallel configurations under varying conditions, including model architectures, parameter sizes, cluster scales, sequence lengths, and global batch sizes, ensuring optimal performance across diverse workloads.

Table 2: Comparison of our FlexParallel and manual expert-tuning performance for multiple models.

| Model | # of NPUs | # Trials/ Search Space | Expert Config | FlexParallel | Speedup |
|---|---|---|---|---|---|
| | | | (PP, TP, DP, CP, UP, VPP, EP, MBS) | | |
| LLaMA-7B | 8 | 6/387 | (8, 1, 1, 1, 1, 1, 1, 4) | (4, 1, 2, 1, 1, 1, 1, 2) | 1.11× |
| LLaMA-13B | 8 | 4/333 | (8, 1, 1, 1, 1, 1, 1, 1) | (8, 1, 1, 1, 1, 5, 1, 1) | 1.02× |
| LLaMA2-13B | 8 | 7/429 | (1, 8, 1, 1, 1, 1, 1, 4) | (1, 8, 1, 1, 1, 1, 1, 4) | 1.0× |
| LLaMA2-34B | 16 | 6/1228 | (2, 8, 1, 1, 1, 1, 1, 2) | (2, 8, 1, 1, 1, 1, 1, 2) | 1.0× |
| LLaMA3-8B | 8 | 5/289 | (1, 8, 1, 1, 1, 1, 1, 2) | (4, 1, 1, 1, 2, 1, 1, 1) | 1.09× |
| GPT-13B | 32 | 9/1040 | (1, 8, 4, 1, 1, 1, 1, 4) | (8, 1, 4, 1, 1, 1, 1, 1) | 1.10× |
| GPT-38B (4K) | 128 | 17/3458 | (4, 8, 1, 4, 1, 1, 1, 1) | (2, 8, 2, 4, 1, 1, 1, 1) | 1.12× |
| GPT-38B (32K) | 64 | 6/2410 | (4, 8, 2, 1, 1, 1, 1, 1) | (2, 8, 4, 1, 1, 1, 1, 2) | 1.07× |
| Qwen2.5-7B | 8 | 5/261 | (2, 4, 1, 1, 1, 1, 1, 1) | (1, 4, 1, 2, 1, 1, 1, 1) | 1.03× |
| Pangu-135B | 6144 | 22/58205 | (8, 8, 96, 1, 1, 6, 1, 1) | (6, 8, 128, 1, 1, 8, 1, 1) | 1.05× |
| Pangu-135B | 8192 | 15/27608 | (8, 8, 128, 1, 1, 6, 1, 1) | (8, 8, 128, 1, 1, 6, 1, 1) | 1.0× |
| mixtral-8x7B | 64 | 31/2314 | (1, 8, 2, 4, 1, 1, 1, 1) | (1, 8, 2, 4, 1, 1, 2, 2) | 1.09× |

## 5 CONCLUSION

Modern LLM training typically employs hybrid parallelism strategies to improve throughput, but identifying the optimal configuration for different models remains a significant challenge. Most existing frameworks either require labor-intensive manual tuning or rely on highly accurate cost models to automate this process. The latter approach is often hampered by the inherent inaccuracies of cost models, resulting in suboptimal parallelism configurations. To address this challenge, we propose FlexParallel, an efficient auto-tuner. FlexParallel employs a probabilistic surrogate model for each operator to predict the duration and its uncertainty. The surrogate model is trained on a combination of simulated performance data and profiling data. An end-to-end cost model is then constructed by combining these surrogate models with domain knowledge of LLM architectures and the implementation details of various parallel techniques. We develop a query-efficient exploration strategy and an early stopping termination mechanism to identify the optimal configuration. Extensive results on 10 mainstream LLMs demonstrate that FlexParallel achieves competitive and even superior overall training performance compared to expert-tuned configurations. Moreover, FlexParallel can identify the optimal configuration with high sample efficiency. To the best of our knowledge, this work is the first empirical validation of an auto-tuning algorithm for distributed training on a massive 8192-device cluster.

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

# A APPENDIX

## A.1 PARALLELISM TECHNIQUE

Multiple strategies have been developed to reduce pipeline bubbles in PP, including interleaved pipeline scheduling (Narayanan et al., 2021), bidirectional pipelines (Li & Hoefler, 2021), model chunk placement (Liu et al., 2023b), and backward computation splitting (Qi et al., 2024). Virtual pipeline parallelism (VPP) (Narayanan et al., 2021) places multiple disjoint model chunks in each device, thereby increasing the number of pipeline stages without requiring additional physical devices. (Qi et al., 2024) effectively reduces the pipeline bubble by dividing the backward pass into two distinct computations and redesigning the microbatch schedule. LLMs with long-context capacities are crucial for tasks such as code generation (Hui et al., 2024; Li et al., 2025), document summarization (Koh et al., 2022), and video understanding (Bai et al., 2025). However, training LLMs with long sequences introduces a memory bottleneck due to the storage of intermediate activations. To address this challenge, sequence parallelism (SP) (Li et al., 2021; Gu et al., 2024; Li et al., 2023) splits input tokens into multiple chunks distributed across devices, with each GPU processing its assigned segment. (Jacobs et al., 2023) proposes the Ulysses Parallelism (UP) approach, which utilizes all-to-all collective communication to aggregate all input tokens before attention computation. (Li et al., 2021) introduces the ring attention method, which enables distributed attention computation through sequential peer-to-peer exchanges of partial key/value matrices across devices. Build upon ring attention, context parallelism (CP) (Nvidia, 2025) enhances SP performance by eliminating redundant computations stemming from the low-triangle structure of causal attention masks. State-of-the-art training systems, such as Megatron-LM (Shoeybi et al., 2019; Narayanan et al., 2021)

and DeepSpeed (Rasley et al., 2020), support the parallelism strategies mentioned above. Manual expert-tuning of parallelization configurations is required for a given model and cluster specification.

## A.2 EFFECTIVENESS OF COST SURROGATE MODEL

Figure 7 reports prediction accuracy of the proposed cost surrogates for three primary operators: matrix multiplication (MatMul), root-mean-square normalization (RmsNorm), and flash attention. In all cases, the uncertainty intervals consistently encompass the ground-truth runtimes, demonstrating the reliability of the uncertainty estimation. At the same time, the prediction is informative: the uncertainty intervals are sufficiently narrow in most scenarios to clearly distinguish performance across input shapes.

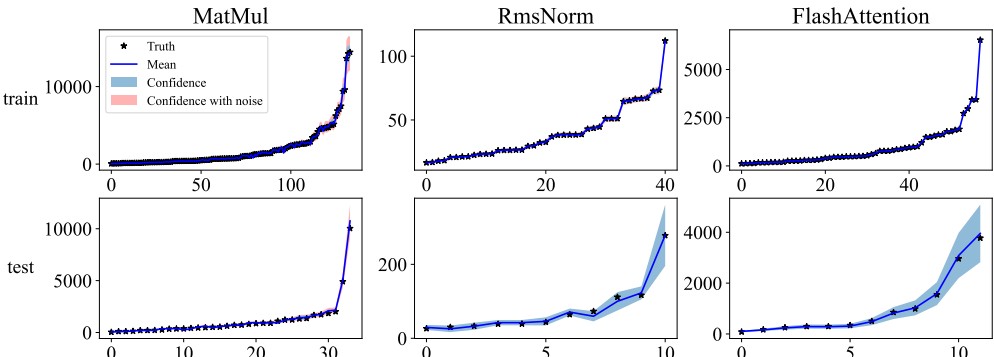

Figure 7: Accuracy of cost surrogates for three operators on training and test sets. The two datasets differ in input shapes. The x-axis is the input-shape index within each set; the y-axis is the predicted or ground-truth runtime (μs).

We evaluated the LLaMA-7B model on an 8-NPU system across a search space of 30 feasible parallelism configurations. Our approach achieves $\tau = 0.84$. In the modeling results of the Calculon method, 6 configurations encountered Out-of-Memory (OOM) errors, including the optimal one. After removing OOM configurations, Figure 8 shows the rank correlation of both approaches. Notably, our cost model achieves $\tau = 0.86$, a significant improvement over the Calculon's baseline $\tau = 0.76$, which underscores its superior rank preservation.

## A.3 LLM USE

We wish to disclose that a large language model was used to assist in the language polishing and proofreading of this manuscript. All scientific content and conclusions are the sole responsibility of the authors.

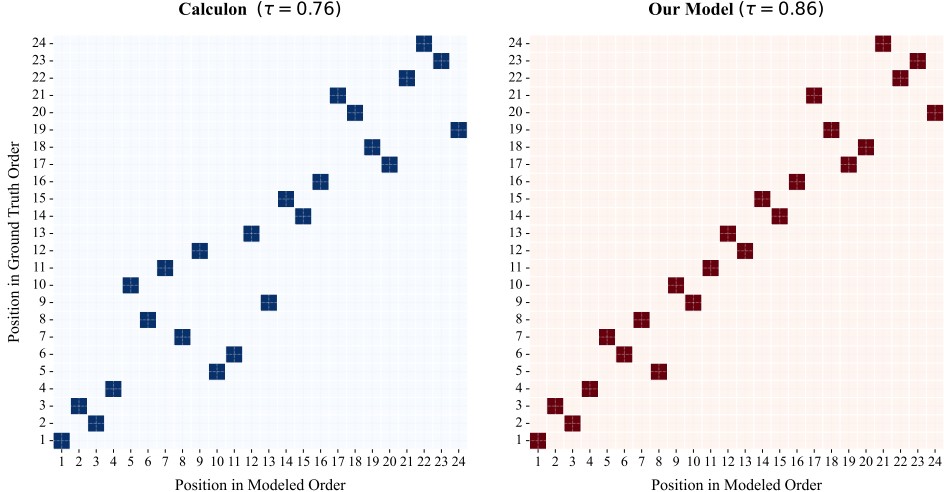

Figure 8: Rank correlation comparison between Our approach and Calculon. A dark-colored cell $(i, j)$ indicates that the parallelism configuration originally at rank position $j$ in the ground truth is positioned at rank $i$ in the modeled output. A perfect diagonal pattern demonstrates exact agreement between predicted and true performance ranking.