# OpenReview forum: "FlexParallel: Automatic Parallelism Tuner via Grey-Box Optimization for Training Giant Models"
_ICLR.cc/2026/Conference — Submitted to ICLR 2026_

### Official Review · Reviewer_j64s · 2025-10-26

**Soundness:** 2
**Presentation:** 2
**Contribution:** 3
**Rating:** 2
**Confidence:** 4

**Summary:**

This paper proposes FlexParallel, which automatically finds the parallelization configuration for training large language models. FlexParallel has three components. First, an uncertainty-aware cost surrogate is initialized with simulated performance data and computes a residual error that serves as the training signal for a Gaussian Process (GP). At prediction time, because the GP models the conditional distribution of the error given the input shape, the GP-predicted residual is added to the initial estimate.
To estimate the cost incurred by a given parallelization configuration and LLM, FlexParallel uses a parallelism explorer based on modified Thompson sampling, designed to avoid revisiting already explored configurations. With this modification, the explorer identifies an efficient configuration with a small number of explorations. The stopping criterion is the probability that the optimal configuration has already been explored; exploration terminates once this probability exceeds a threshold.
The search space includes seven parallelism methods (TP/DP/PP/VPP/CP/UP/EP). The authors evaluate FlexParallel on various model architectures—LLaMA, GPT, Qwen2.5, PanGu, Mixtral (MoE)—and scale up to 8,192 NPUs. With the proposed methods, FlexParallel achieves up to 1.12× speedup over an expert-tuned configuration.

**Strengths:**

- This work targets the practical problem of finding the optimal configuration for LLM training parallelization.

- It adopts a Gaussian Process to model the residual error for estimation, which aids analysis and interpretability.

- It tests diverse model architectures and includes various parallelization techniques (PP/TP/DP/CP/UP/VPP/EP) in its search space.

- It also considers micro-batch size (MBS) when searching for the configuration—important because techniques like PP and VPP depend on MBS, which affects training speed.

- It evaluates on a large number of NPUs and shows effectiveness at scale (e.g., 6,144 NPUs). For PanGu-135B with 6,144 NPUs, the method achieves a 1.05× speedup over an expert-tuned configuration.

**Weaknesses:**

- My main concern is the lack of strong baselines. Although the paper claims speedups over an expert-tuned configuration, it is unclear who the “expert” is and how widely that setup is adopted. Moreover, there are existing systems for automatic configuration (e.g., AMP [1], Pipette [2]); including these baselines and demonstrating superior performance is necessary.

[1] D. Li, H. Wang, E. Xing, and H. Zhang, “Amp: Automatically finding model parallel strategies with heterogeneity awareness,” Advances in Neural Information Processing Systems, vol. 35, 2022.


[2] J. Yim, J. Song, Y. Choi, J. Lee, J. Jung, H. Jang, and J. Lee, “Pipette: Automatic fine-grained large language model training configurator for real-world clusters,” in Proc. 2024 Design, Automation & Test in Europe Conf. & Exhibition (DATE), 2024.

- I am also concerned about reproducibility. The work uses a large number of NPUs, which may be difficult to access. Without open-sourcing the code, reproducing the results will be challenging.

- The transparency of the experimental setup is insufficient. The paper does not clearly state the network bandwidth between nodes, nor the intra-node interconnect bandwidth.

- The reported speedups are somewhat marginal. In Table 2, many configurations achieve less than 1.1× speedup, and some show no speedup at all.

- A key omission is the actual wall-clock time per training iteration. The paper reports only speedup ratios and does not include absolute iteration times.

- The paper does not validate the GP assumptions (jointly Gaussian function values and Gaussian observation noise). Please check and report whether these assumptions hold for the operator-level error e given input x.

**Questions:**

- Baselines: Who is the “expert” behind the expert-tuned configuration, and how common is that setup? Can you include stronger automatic-configuration baselines (e.g., AMP [1], Pipette [2]) and show superior performance?

- Reproducibility: Will you open-source code, configs, and scripts so that results—especially those requiring large NPU counts—are reproducible?

- System details: What are the exact networking specs (inter-node bandwidth and intra-node interconnect bandwidth) used in the experiments?

- Wall-clock time: Can you report absolute wall-clock iteration times (not just speedup ratios) for representative settings?

- GP assumptions: Have you validated the GP assumptions (jointly Gaussian function values and Gaussian observation noise) for the operator-level error e given input x?

**Details Of Ethics Concerns:**

This work does not raise any specific ethical concerns.

---

### Official Review · Reviewer_u4CY · 2025-10-28

**Soundness:** 2
**Presentation:** 2
**Contribution:** 2
**Rating:** 2
**Confidence:** 3

**Summary:**

This paper proposes FlexParallel, a grey-box automatic parallelism tuner that learns to predict the runtime of individual operators under different parallel configurations and uses Bayesian optimization to find an efficient configuration for large-scale training. The system builds a Gaussian-process-based surrogate model that estimates operator-level runtime distributions conditioned on input shape and parallel strategy, then aggregates the predicted costs across operators to approximate the total training time. Experiments on large-scale Ascend NPU clusters (up to 8192 devices) show up to 1.12× speedup compared with an “expert configuration”.

**Strengths:**

The paper tackles an important and practically relevant problem—efficient automatic parallelism tuning for large-scale model training.
The idea of combining a grey-box cost model with uncertainty-aware exploration is sound and aligns with recent research trends.
Demonstrating experiments at the 8K-device scale is impressive in terms of engineering implementation.

**Weaknesses:**

The technical contribution and empirical validation are not convincing.
First, the cost modeling is incomplete: the surrogate predicts only the runtime of individual operators under a given configuration, but communication overhead is entirely ignored. In distributed training, communication time (e.g., all-reduce, all-to-all) often dominates runtime and is tightly coupled with parallel strategy. Neglecting this part makes the optimization target unreliable.

Second, the model’s prediction error is large and unbounded. Figure 3 shows that even for a single MatMul operator, relative errors range from −0.2 to +0.8 across input shapes. When aggregated over many heterogeneous operators, the accumulated error can become substantial, yet the paper provides no discussion or sensitivity analysis on its impact at the model level.

Third, there is no analysis of prediction accuracy across parallel configurations. The paper does not evaluate how the surrogate performs for unseen configurations or different parallel dimensions (data, tensor, pipeline, expert). Without such analysis, it is unclear whether the surrogate can generalize beyond the measured samples.

Fourth, the experimental setup lacks transparency. The so-called “expert configuration” used for comparison is not clearly defined or justified; it is unclear whether it comes from human tuning, vendor heuristics, or prior systems. The claimed improvements are marginal—around 12% at best—and almost vanish when scaling from 6144 to 8192 NPUs. This weak scalability suggests that the system’s optimization effect is minimal under realistic cluster sizes.

Fifth, key details are missing: the paper does not report the number of sampled configurations, per-trial overhead, or total tuning time. Without these, it is impossible to judge the real benefit or efficiency of the proposed method.

Overall, the paper reads more like an engineering prototype than a rigorously validated research contribution. The absence of communication modeling, large surrogate error, incomplete analysis, and limited speedup make the work technically unsound and empirically unconvincing.

**Questions:**

- How is communication (e.g., all-reduce, all-to-all) modeled or measured in the cost function? If it is ignored, how can the model capture scaling behavior?
- The relative error for individual operators is high. How does this error propagate to the total runtime prediction?
- What is the prediction accuracy of the surrogate under different parallel configurations and unseen shapes?
- How was the “expert configuration” baseline obtained? Is it tuned by hand or based on a known framework?
- What is the total number of samples and tuning trials required before convergence? What is the runtime cost of each trial?

---

### Official Review · Reviewer_RxAm · 2025-10-29

**Soundness:** 2
**Presentation:** 1
**Contribution:** 2
**Rating:** 2
**Confidence:** 4

**Summary:**

The paper introduces a method to find the best parallelism configuration to reduce the per-iteration runtime. This is done by using a cost model   along with measured runtime of each operators to construct a surrogate model at the operator level. With the surrogate models, the best configuration is chosen via thompson sampling, and early stopping is performed to stop the search when a good configuration has been found. The paper demonstrates that the method improves on expert selection of parallelism configuration on multi-NPU systems.

**Strengths:**

The targeted problem is practical (due to the interest in large model training), and shows a different viewpoint for the area of tuning parallelism configurations.

There is also an attempt to propose a method (GP + surrogate model) to solve this issue, which have not been considered by current literature.

Experiments show good results for training scenarios with a large number of NPUs. This demonstrates the method can potentially scale well on multi-NPU systems.

**Weaknesses:**

Several key benchmarks for comparison and ablations are missing, including:

- Existing frameworks like Nemo and Deepspeed have methods to perform tuning for parallelism configurations (e.g., the Autotuner in Deepspeed). These methods work under the same ethos of using actual measurements to inform which parallelism configurations are good, however are not compared with by the paper.

- Other methods like Alpa, NNScaler, AMP, etc., which are based on cost model that the author have mentioned should also be compared against. While the authors acknowledge these methods and their limitations, it should still be ran to provide some baselines of other methods which are standard within this field.

- Related, the author seems to also run experiments to compare the methods with Calculon -- why can this algorithm not be used for selection as a comparison also (by treating it solely as a cost model to get the single maximum performing configuration)? In Figure 8, since both methods is able to identify the best configuration correctly (position 1 chosen correctly in both cases), it would suggest that using Calculon alone without the GP might work as well.

- Simple ablation such as the proposed method without the GP modelling (but just the cost model alone), or method without the early stopping are not compared with. It is therefore unclear whether the proposed changes are actually beneficial to the algorithm at all.

Meanwhile, the only benchmark reported is expert selection, which is not a good baseline since it can be subjective by the expert as to what is a good configuration (I might have just asked a "bad" expert who gives suboptimal configuration in the first place). There is little transparency here with this method due to the lack of algorithmic clarity.

While the paper demonstrates large scale experiments, other settings may be considered as well:

- The authors have mainly considered LLMs of different sizes, however have limited their experiments to language models. It would be interesting to see how the method compares on other workloads as well.

- The experiments are also limited on training specifically on NPUs (with the exact configuration also unclear to whether they are single-host or multi-host and how many hosts they are distributed on). This fact seems to not be mentioned until the experiments section. It would be better to show how the method works on other setups as well, such as on systems with GPUs (which are more commonplace in practice) on single- and multi-host setups across larger variations of connectivities as well.

Several design choices of the algorithm could be elaborated better:

- The authors constructed surrogate models for the cost of each operations, with a short justification that it can be reused in different tasks. This claim is not explained well, and no experiments demonstrate the ability to reuse these values. Additionally, there are other ways a surrogate can be constructed (e.g., modelling the end-to-end timing which may also incorporate overheads overlooked by individual operations) -- it would be nice to elaborate why these approaches were not taken.

- The authors use a GP to model the relative error score. It is unclear as to why a GP is used in this case (I suspect it is related to the uncertainty being captured by the GP but this is not clear from the text itself). Why is the GP used to capture the relative error instead of some of the other quantities? This seems to be a design choice that should be mentioned more since it may not be obvious as to why it is used.

- The authors mention the possible use of UCB, however chooses to not use so because of how it has no closed form. However, this is not necessarily a real issue, since UCB can be computed through monte carlo methods as done in some black-box optimisation packages in practice, and that some of the other criteria used in practice for these problems do not have closed forms anyway (including thompson sampling which relies on sampling too). Better reasoning would therefore be useful here.

- Early stopping for black-box optimisation methods already exist, which isn't explored by the authors. It would therefore be good for a clearer comparison with these methods, and as to why the proposed method is better than these ones (if they are at all different). Most convincing way to do so would be through showing how the regret bound changes with the proposed methods by the paper.

Several aspects of the algorithm is omitted, which should be included in the paper (at least in the appendix) such that it is more self-complete. For example, how is the cost exactly computed in this case, with what forms of f_i and g_i for some of the architectures the paper formulated and defined? How are the GPs formulated, and with what hyperparameters (or if they are chosen, how are they chosen)? In Line 353, what are the acronyms referring to? (they are mentioned in the appendix but can be missed by readers as well).

None of the results report error bars or variance across multiple runs. It is difficult to compare the statistical significance of the results without these information, especially when some of the speedup reported is relatively small compared to the benchmark. If I ran the Thompson sampling with a different random seed, could I have ended up with vastly different results?

**Questions:**

Some of the issues I have are mentioned in the Weaknesses section. Other ones I have are as follows:

1. The algorithm seems to simply use prior knowledge and combine with thompson sampling and early stopping methods, all which are already commonplace in existing black-box/grey-box optimisation methods. In this case, what novelties are the authors proposing in the work beyond application of existing techniques on this problem? It would be appreciated if the motivations for this bit is explained better in the text.

2. What is the runtime of the algorithm? Would it significantly take up much more time from the training, especially compared to existing methods to tune the parallelism configurations? It is difficult to make a statement about the efficiency when it doesn't show a comparison that is more efficient with respect to something.

3. Why do the authors limit their work to the hyperparameters they have chosen? For example, there are other aspects for PP that may be controlled as well, like how many pipeline stages fit on the same GPU in some interleaving fashion. Could the algorithm be easily extended to also include these factors?

4. How much does the algorithm rely on an accurate cost model? If the cost model is misspecified or comes across as system that was not modelled well, would the algorithm still converge as well or recover a good parallelism configuration as currently done?

---

### Official Review · Reviewer_wu2H · 2025-11-05

**Soundness:** 2
**Presentation:** 1
**Contribution:** 2
**Rating:** 4
**Confidence:** 2

**Summary:**

The paper introduces FlexParallel, a framework for automatically finding the optimal hybrid parallelism configuration for training LLM. The main idea of the approach is to explore different configurations using upper confidence bound (UCB). The method is validated in large-scale clusters of NPU and demonstrate improvements over the manually tuned parallelism configuration.

**Strengths:**

* FlexParallel scales up to 8192 devices, demonstrating its effectiveness of truly large-scale training.
* The method is efficient in its exploration and only requires exploring negligible fraction (2%) of the whole search space.
* The method is applicable to general LLM architectures.

**Weaknesses:**

* The function $u$ (Equation 1), which synthesizes the end-to-end runtime from individual operator costs, is described as an "explicit rule". However, the paper does not detail the precise logic of this rule, particularly how it models the intricate interdependencies of various hybrid parallelism methods.
* All empirical validation was conducted on Ascend clusters using the MindSpeed framework. While the concept is transferable, the specific performance gains and optimal configurations found are likely hardware-specific, which may limit the direct applicability of the results to other common platforms like NVIDIA GPUs.
* The modified Thompson sampling is explicitly designed to minimize the total tuning time by avoiding re-exploration of already-tried configurations. While justifiable due to high switching overhead, this deviates from standard Thompson sampling's goal of minimizing long-term cumulative cost. The design choice is a highly heuristic; its long-term optimality guarantees are not explicitly discussed.
* The choice of expert configuration is rather arbitrary, and the acceleration is a bit marginal.

**Questions:**

* Could authors provide a concrete example on all the terms appeared in Equation 1, i.e. $u$, $g_i$, config, etc.

---

### Meta-Review · Area_Chair_xrzk · 2026-01-13

**Summary:**

FlexParallel proposes using Gaussian Process-based Bayesian optimization with Thompson sampling to automatically tune parallelism configurations for LLM training on large clusters (up to 8192 NPUs). While the problem is practically important, the paper has fundamental technical and experimental shortcomings. Most critically, the cost model ignores communication overhead (all-reduce, all-to-all), which often dominates runtime in distributed training and is tightly coupled with parallel strategy—this makes the optimization target unreliable. The method section lacks clarity on how Equation 1 synthesizes end-to-end runtime from operator costs. The baselines are weak: the "expert configuration" is arbitrary and not well-defined, and no comparison is made with existing automatic tuning methods (DeepSpeed Autotuner, Alpa, AMP, Pipette). The surrogate model shows large prediction errors (−0.2 to +0.8) without analysis of error propagation. Missing details include number of samples, tuning time, network bandwidth, error bars, and ablations.

**Reviewer Concerns:**

No author rebuttal was provided. All concerns remain outstanding: (1) Communication overhead ignored in cost model—fundamental flaw; (2) No comparison with existing automatic tuning methods (DeepSpeed Autotuner, Alpa, AMP, Pipette); (3) "Expert configuration" baseline undefined and arbitrary; (4) Large surrogate prediction errors without propagation analysis; (5) Missing ablations (without GP, without early stopping); (6) No error bars or variance across runs; (7) Hardware-specific (Ascend NPUs only), no GPU validation; (8) Missing details (samples, tuning time, bandwidth, GP hyperparameters); (9) No wall-clock times, only speedup ratios; (10) GP assumptions not validated; (11) Reproducibility concerns with no code release.

**Reviewer Scores:**

No author rebuttal was provided, scores likely unchanged.

---

### Decision · Program_Chairs · 2026-01-26

Reject